# Launch Performance Degradation of the Rupture-Type Missile Canister

**Wonhong Choi [1] and Sunghun Jung [2],***

[1] Department of Mechanical Engineering, Purdue University, West Lafayette, IN 47907, USA; choi124@purdue.edu
[2] Department of Electric Vehicle Engineering, Dongshin University, Najusi, Jeonnam 58245, Korea
* Correspondence: jungx148@dsu.ac.kr

**Abstract:** This paper describes the degradation of launch performance caused by the remnants of a missile canister cover with a sabot interface, on interference with adjacent structures. First, by including the material plastic behavior and element deletion, we predict interference between the structures and the detached part, followed by excessive deformation. Second, we verify that the support ring deformation, which is induced by an interaction with the cover remains, increases for fastener separations with abnormal fastener installations. This increase further triggers interference with the boosters on the bottom of a missile. Lastly, we analyze the variation of material property in a high-speed environment.

**Keywords:** finite element analysis (FEA); impact behavior; missile canister

## 1. Introduction

A missile canister protects the missile from foreign matter inflow, shock vibration, and weather such as rain and snow, and facilitates smooth ejection at launch. The typically preferred canister cover is the rupture-type, which is easily manufactured and gives excellent handling convenience. However, if the missile canister cover fails to rupture at a missile launch, it could possibly interfere with the adjacent structure.

The rupture-type canister cover is circular in shape and is installed with a sabot between the missile and canister to guide the missile during launch. The sabot is made of urethane foam and restrains displacement in the radial direction. Springs installed between the canister and missile are broken away with the missile, thrown out, and finally, they fall. The missile canister cover, which is also made of urethane foam, is ruptured by the propellant ignition impact pressure formed inside the missile canister and is removed.

Cover remnant analysis: Kim, R. et al. [1] originally devised the rupture-type canister cover through their patent, confirming that the reliability of the cover opening would be increased by applying notches on the surface. Chung, J. et al. [2] performed the first structural analysis and validation test for the rupture-type canister cover. Yang, Y. [3] demonstrated the mechanism for the cover to open by internal pressure propagation in the missile canister through computational fluid dynamics. Yang, Y. et al. [4] described the behavior of the rupture-type cover opening through a comparison of a field test with a real rocket motor. However, the aforementioned studies did not recognize that cover remnants caused by improper manufacturing or insufficient internal pressure build-up may affect other components inside the canister owing to mutual collisions or contact with each other.

Material deterioration analysis: with respect to the material deterioration of the urethane foam associated with the parametric study presented below, several foregoing studies are discussed. Vries [5] demonstrated that the general form of the strain–stress curve for polymeric foam has both plateau and

densification processes, unlike metals. The report of the Argonne National Laboratory [6] states that the degradation of urethane foam may be related to the following causes: temperature effect, water immersion, biodegradation, and ultraviolet radiation. Among these, the temperature and radiation effects were believed to be the main contributors to the weakening strength and stress with long-term storage. At 90 °C, the compressive strength of the material dropped to half owing to the softening of the polymers and damage caused by radiation, which is believed to not be critical.

Material property variation in high-speed analysis: the dynamic behavior of the material at high speed was investigated in several studies. In general, the Johnson–Cook constitutive model was introduced to reflect the strain rate effect on the strain-stress curve under a high-speed environment [7].

Problem statement: to quantitatively analyze the effect of the presence of remnants on the ejecting performance of the missile in terms of structure, we conducted an impact analysis based on a commercial finite element analysis (FEA) tool. In several field tests, we observed that cover debris may exist due to improper manufacturing and lack of ignition pressure, and there has been almost no consideration of a scenario where the cover debris is not entirely removed after ignition. In a certain flight test, the missile failed to launch as the ring structure of the canister interfered with the rear wing in the middle of lift-off. We hypothesized that the ring structure might be detached by a certain contact-induced force based on the cover debris. In this regard, we conducted an FEA-based impact analysis to quantitatively analyze the cause of the failed event. To the best of our knowledge, very few studies have been performed to investigate the exact causes of the degradation of the launch performance.

The remainder of this paper is structured as follows. Section 2 presents the overall structure of the rupture-type missile canister. Section 3 explains the three sources of launch performance degradation. Section 4 provides the test environment and Section 5 shows corresponding test results. The concluding remarks are given in Section 6.

## 2. Structure of the Rupture-Type Missile Canister

Figure 1 shows the missile canister body made of a cylindrical aluminum alloy (AL5083-H112), with the cover assembly mounted on both the front and rear sides [8]. Two stacking frames for lifting and loading purposes are installed around the canister.

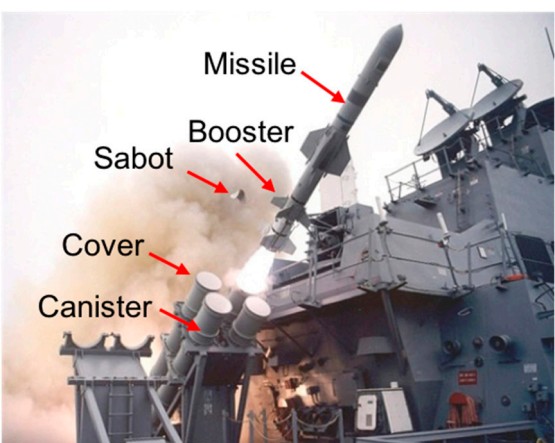

**Figure 1.** Overview of a similar canister for a Harpoon missile.

The assembly structure of the missile canister cover is shown in Figure 2. The rupture cover is made of a urethane foam material and has cross-shaped notches to facilitate rupture. The rupture-type cover is inserted into a missile canister frame, attached using a silicone adhesive, and is secured by a support ring (SUS304) that is fastened with 16 dish head fasteners. The interior of the canister contains the missile and three guiding sabots spaced 120° apart. Studs are installed between the missile and

sabot to normally guide the missile during ejection. A spring repulsive force makes the stud move away from the missile after ejection.

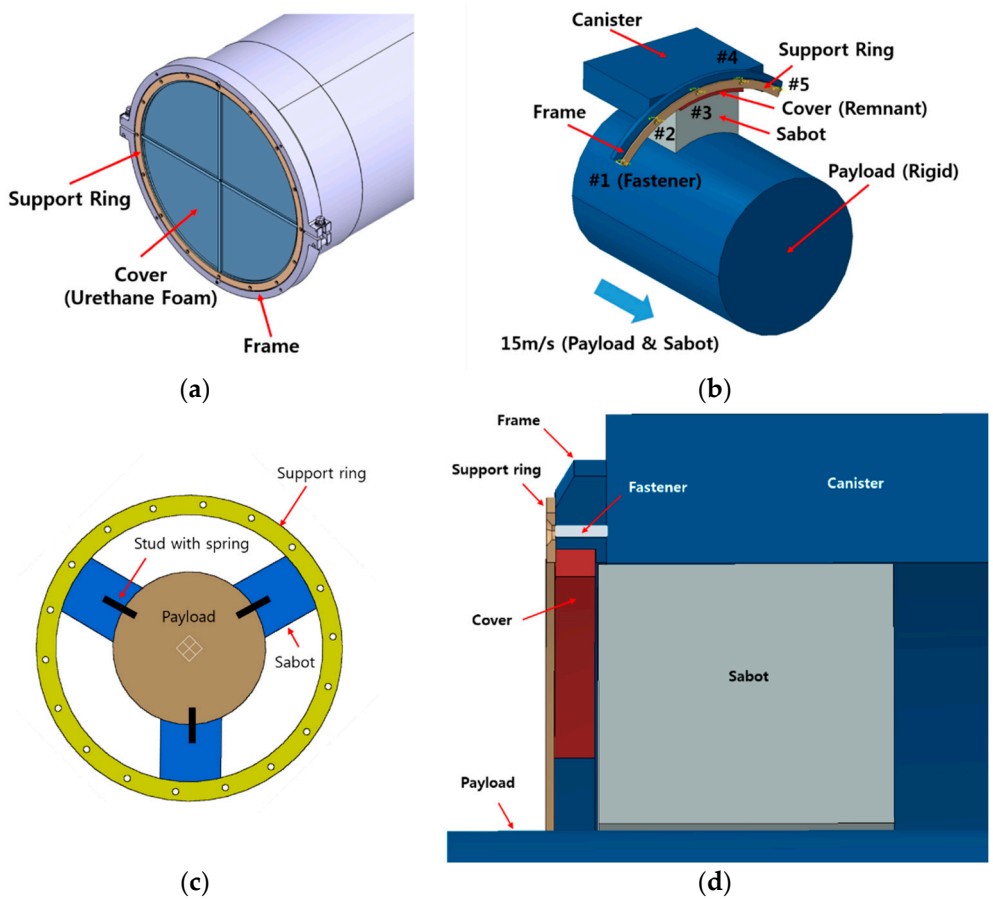

**Figure 2.** Missile canister cover assembly: (**a**) external isometric view, (**b**) internal isometric view, (**c**) section view 1, and (**d**) section view 2.

Sometimes during launch, the canister cover does not completely rupture, and residues are generated, which interfere with the moving sabots, and the support ring of the missile canister cover is torn off. As a result, the support ring gets caught in the booster at the lower part of the missile, which prevents the booster blade from unfolding. When the missile canister cover is not completely removed by the ignition shock wave before launch, the cover remnants are left between the support ring and canister and it eventually presses and fixes the rupture-type canister cover.

The sabot moves and exerts a force on the support ring through the cover remnant. As a result, when the support ring is deformed, and the fastening screws cannot hold, the probability of the support ring interfering with the missile booster is increased, and then the support ring is ultimately detached from the frame.

## 3. Three Sources of Launch Performance Degradation

### 3.1. Support Ring Deflection Due to the Presence of Cover Remnant

The deflection of the annular support ring is calculated using the transverse deflection of an annular circular plate subjected to a concentrated load:

$$w = \frac{k_w P}{Drr} \tag{1}$$

$$D_{rr} = \frac{Et^3}{12(1 - v^2)} \tag{2}$$

where $w$ is the deflection (m), $k_w$ is the deflection coefficient (m$^2$), $P$ is the concentrated force acting on the annular ring (N), $D_{rr}$ is the uniform flexural rigidity (Nm), $E$ is Young's modulus (GPa), $v$ is Poisson's ratio (No Unit), and $t$ is the thickness of the ring (m) [9].

*3.2. Material Deterioration in the Urethane Form*

Tcharkhtchi [10] investigated the effect of thermo-aging on the urethane foam using Young's modulus and the maximum stress of tensile tests based on the chemical reaction after 62 weeks of aging at 85 °C. The experimental results showed that both Young's modulus and the maximum stress were decreased by more than half. Based on these results, we assumed that the degradation of the urethane foam could result from long-term storage.

*3.3. Material Property Variation of the Urethane Form at High-Speed*

It is unfortunately difficult to find test examples at high speed, such as hyper-foam materials, e.g., urethane-foam. Thus, as an alternative, we referred to [11–13] in which dynamic tensile tests for high strength steel were conducted and retrieved some insights into predicting the material behavior of urethane-foam at a high speed of 15 m/s. Based on their results at 15 m/s, Young's modulus, which is the initial slope of the stress–strain curve, is more than doubled, while the tensile strength was increased by up to 20%. From this approximation, we assumed that the urethane foam property at high speed occurs at the point where Young's modulus and the tensile strength are doubled.

## 4. Test Environment

We conducted an impact analysis using the FEA tool to quantitatively identify the probable cause of the missile booster of the support ring being caught at the launch process. To simplify the analysis, we performed a geometrical simplification and employed the commercial Abaqus 6.14-1 FEA tool.

*4.1. Modeling*

A simulation, which reflects the actual shape of the missile canister system, was carried out for the impact analysis. For the simplification of the analysis, the missile was assumed to be a rigid body, and only the quarter symmetrical part was analyzed.

The components and properties of the materials are shown in Tables 1 and 2, respectively. Here, the properties of the urethane foam were obtained from a tensile/bending test (ASTM D790-97), reflecting the plastic deformation behavior. The plastic deformation behavior of the support ring material, SUS304, was obtained using the Johnson–Cook Plasticity Model [7]. For the missile canister and frame material, AL5083, only the elastic region was considered owing to its small deformation.

**Table 1.** Material component.

| Component | Material |
| --- | --- |
| Cover | Urethane Foam 400 |
| Sabot | Urethane Foam 450 |
| Support Ring | SUS304 |
| Frame | AL5083-H112 |
| Canister | AL5083-H112 |

**Table 2.** Material property.

| Factor | UF400 | UF450 | SUS304 | AL5083-H112 |
|---|---|---|---|---|
| Density, $\sigma$ [kg/m$^3$] | 400 | 450 | 8000 | 2700 |
| Young's Modulus, $E$ [GPa] | 286 | 346 | 193 | 70 |
| Poisson's Ratio, $\nu$ [No Unit] | 0.3 | 0.3 | 0.29 | 0.33 |
| Yield Strength, $\sigma_y$ [MPa] | 9.8 | 11.6 | 215 | 190 |
| Tensile Strength, $\sigma_t$ [MPa] | 13.3 | 15.8 | 505 | 300 |
| Plasticity | Yes | Yes | Yes | No |

Regarding the size of the missile canister cover remnants, their width was set to be larger than that of the missile sabot, while the height was set to the maximum measurable value of 5 cm. In addition, the five clamping parts were modeled as a three-dimensional cylindrical cylinder with an effective diameter, and the fastening part of the missile canister is constrained using the multi points constraints (MPC) beam element (1D rigid beam).

### 4.2. Boundary and Load Conditions

The missile and sabots were restrained by the studs while moving inside the missile canister at an assumed velocity of 15 m/s. To shorten the analysis time, we analyzed the missile sabot from the point adjacent to the residue. The analysis was dynamic and explicit and was conducted with a span of 20 ms.

The missile canister and its frame were constrained by a translational three degrees of freedom (DOF). The end of the fastening part was also constrained as a fixed point. We employed the general contact function, which is a default option in Abaqus 6.14-1, to set the type of contact between components.

We also employed a special feature of element deletion for the Urethane foam supported by Abaqus 6.14-1, in which the element was removed at a plastic strain of 6% or greater, to analyze the fracture behavior at the contact surface. The spring at the bottom of the missile sabot was omitted in this numerical model.

## 5. Test Result

Figure 3 shows the time-lapse release pattern of the missile obtained when the clamping part works normally. After pitch-up of the missile sabot, the edges were damaged by the cover remnants, and then the missile sabot breaks the cover remnants. Subsequently, the missile sabot passes through the missile canister and the residue was scattered.

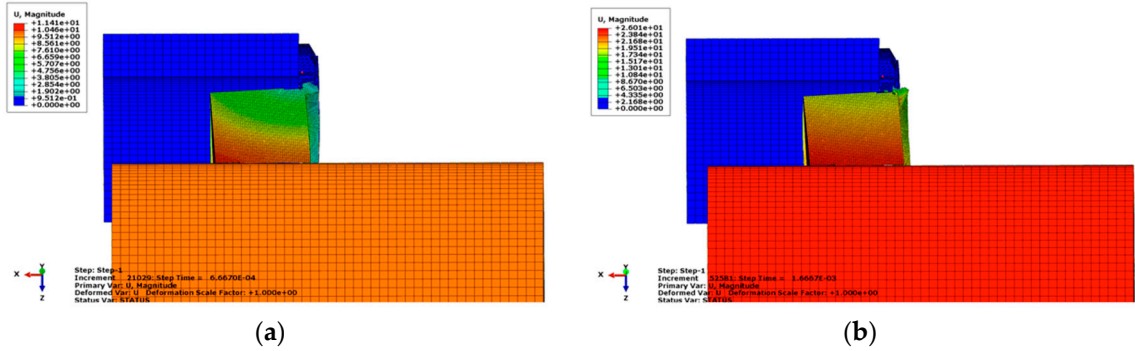

(**a**)       (**b**)

**Figure 3.** *Cont.*

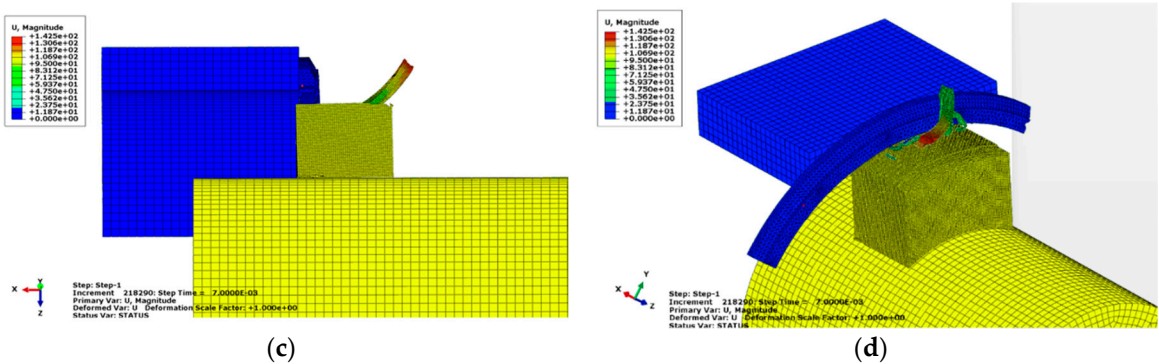

**(c)** **(d)**

**Figure 3.** Analysis results (fastener stable): (**a**) beginning, (**b**) middle, (**c**) end, and (**d**) isometric view at the end.

*5.1. Cover Remnant Analysis*

5.1.1. Cover Remnant Analysis: Deflection of Support Ring

By assuming the support ring is simply supported along with bolt assembly line, and that the contact force acts on the central line between two adjacent bolt points (Figure 4), the concentrated force $P$ can be evaluated from the average value of two reaction forces at points #3 and #4 in Figure 2b, calculated by numerical analysis. In Adewale's study [9], it was observed that the deflection coefficient varies with the ratio of the inner radius ($c$) to the clamped radius ($a$), and circumferential angle ($\theta$). We chose the minimum value 0.0002 m$^2$, assuming that $c/a > 0.7$ and $t = 0.0032$ m. The variables associated with this calculation are summarized in Table 3.

**Table 3.** Verification of the support ring deflection.

| Factor | Value |
|---|---|
| Concentrated force, $P$ [N] | 3500 |
| Reaction force, $R_{max}$ [N] | 3747 |
| Reaction force, $R_{min}$ [N] | 3253 |
| Young's Modulus, [GPa] | 193 |
| Poisson's Ratio, $\nu$ [No Unit] | 0.29 |
| Thickness, $t$ [m] | 0.0032 |
| Flexural rigidity of plate, $D_{rr}$ [Nm] | 575.4 |
| Deflection coefficient, $k_w$ [m$^2$] | 0.0002 |
| Deflection, $w$ [m] | 0.0012 |

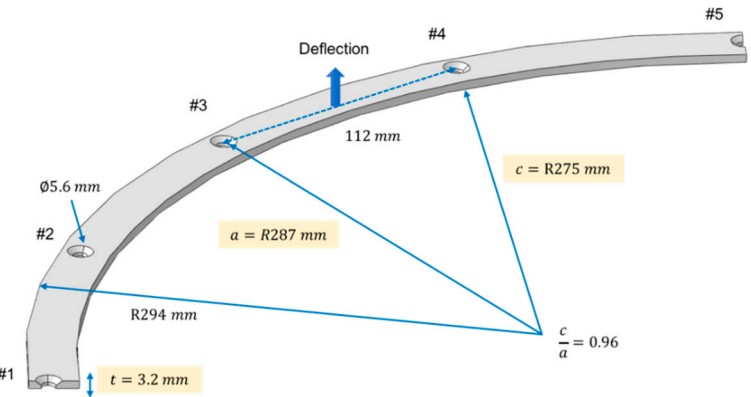

**Figure 4.** Geometry of the support ring.

In this case, the plastic deformation could occur in the support ring and we verified this through the simulation and experiment results, as shown in Figure 5.

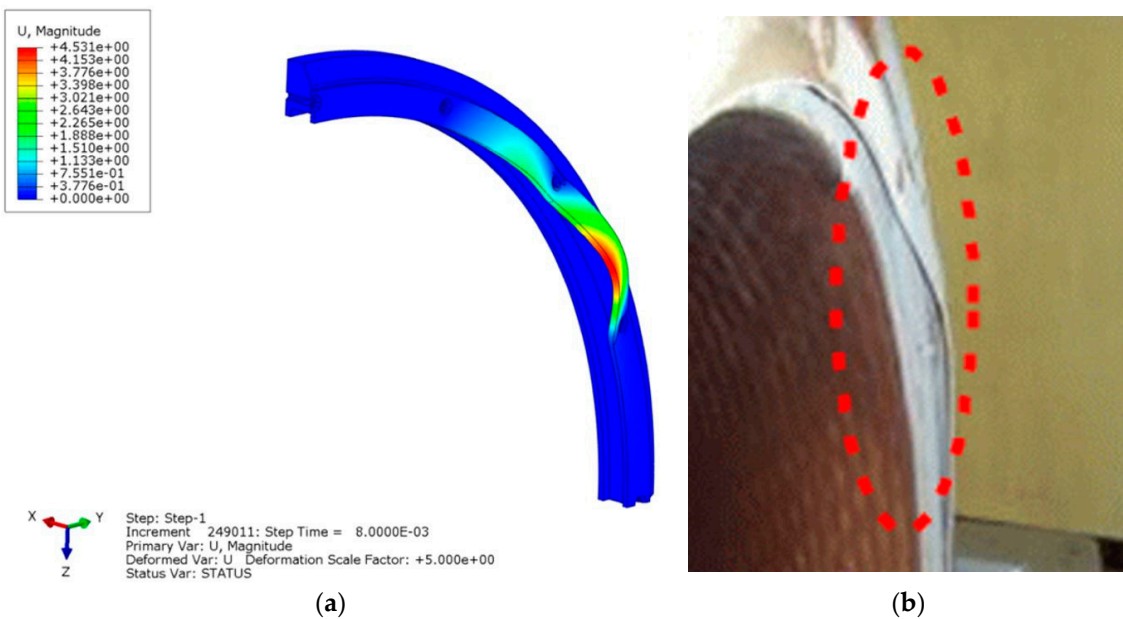

**Figure 5.** Plastic deformation on the support ring (fastener stable): (**a**) computer-aided engineering (CAE) SW and (**b**) visual inspection.

The measured deflection of the actual support ring was 1.2 mm (Figure 5b), which is four times less than the simulation result of 4.6 mm (Figure 6). Since the ring is not fully supported in the simulation, the deflection result of the simulation result should be larger. Nevertheless, in terms of magnitude the two results are very similar. Therefore, we conclude that an increase in the number of bolts may reduce the deflection as it tends to be simply supported and hence, the possibility of detachment of the ring is reduced.

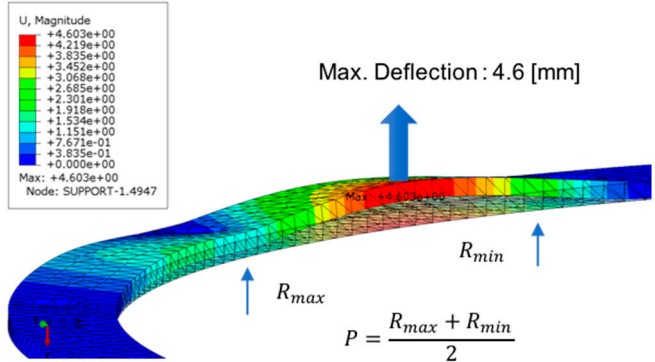

**Figure 6.** Deflection of the support ring (fastener stable).

### 5.1.2. Cover Remnant Analysis: Effect of Cover Residue Height

In this section, we examine the variation of the height of the cover residue. Although the height of the cover residue was approximately less than 5 cm by actual measurement, we assumed the worst-case scenario and set it to 20 cm under normal fastening conditions.

Based on the simulation results, as shown in Figure 7, the effects of the cover remnants on the missile sabot and support ring were found to be insignificant. The cover remnant with an approximate groove height was equally cut by the support ring. It was assumed that the portion protruding beyond the groove height did not significantly affect the deformation of the support ring or the behavior of the missile sabot. The measured axial force formed in the fastening part was 382 kgf.

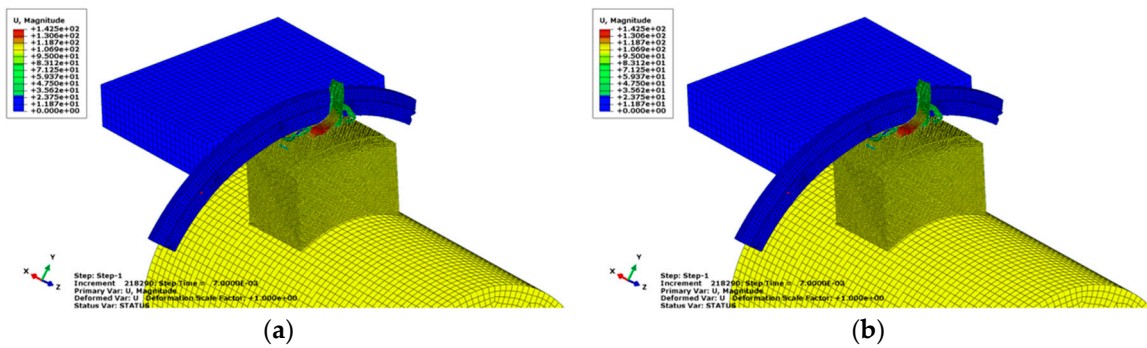

**Figure 7.** Effect of the cover remnant thickness (fastener stable): (**a**) 5 cm and (**b**) 20 cm.

### 5.1.3. Cover Remnant Analysis: Deflection of Support Ring under the Elimination of Fasteners

When the missile sabot was in contact with the cover residue, the missile sabot increased its pitch angle and a part of the edge was damaged, as shown in Figure 8.

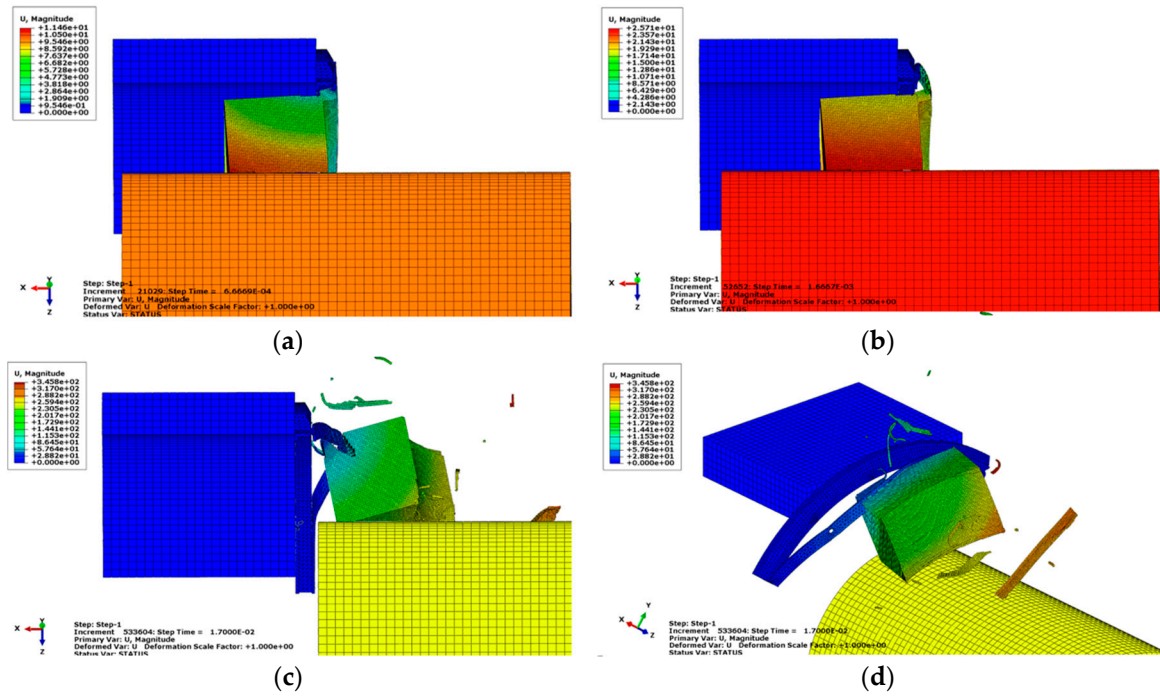

**Figure 8.** Analysis results (fastener unstable): (**a**) beginning, (**b**) middle, (**c**) end, and (**d**) isometric view at the end.

After the collision of the missile sabot and the residues, the former caused a displacement of the missile support ring. The support ring was then inclined toward the sabot that eventually broke off. Finally, plastic deformation occurred in the support ring, as shown in Figure 9. It can be concluded that the deformed support ring might have interfered with the missile booster wing located outside the missile during release.

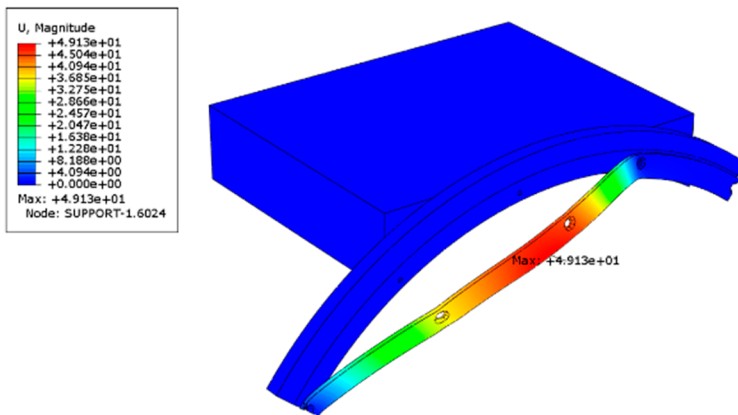

**Figure 9.** Plastic deformation on the support ring (fastener unstable).

### 5.2. Material Deterioration Analysis

To simulate urethane deterioration due to long-term storage, which may cause severe problems, we performed a structural analysis under the assumption that the fasteners were in a normal condition. Urethane foam becomes hardened and brittle when stored for a long time. In the analysis, we doubled the urethane stiffness and reduced the fracture strength to half, as described in Sections 1 and 3.2. As shown in Figure 10, the brittleness increased when the axial force was a maximum. At the end of the simulation, we noticed an increased number of broken pieces.

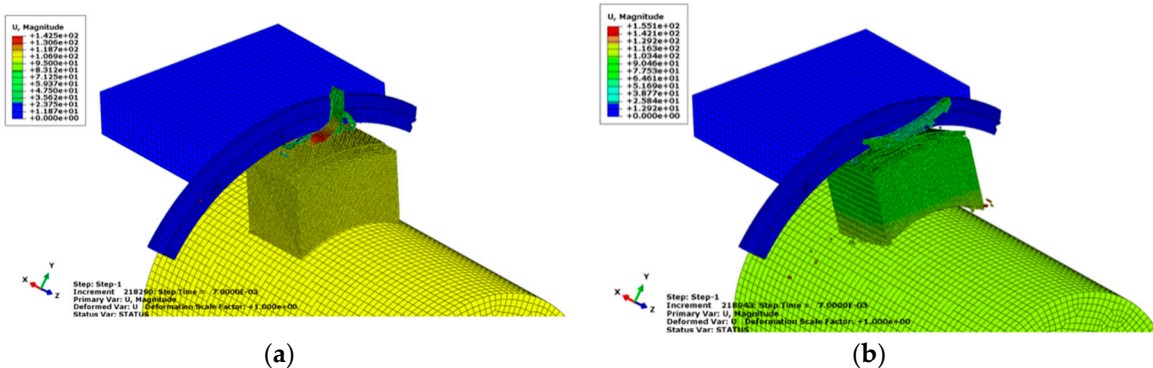

|  |  |
|:---:|:---:|
| (**a**) | (**b**) |

**Figure 10.** Effect of the urethane foam degradation (fastener stable): (**a**) normal and (**b**) degradation.

Based on the simulation results, the maximum axial force was reduced from 382 to 295 kgf. Moreover, owing to the premature failure of the cover residue that acted as a load transferring medium, the amount of support ring deformation and axial force generated in the fastening part were likewise reduced. This provided confirmation that urethane deterioration does not cause a fundamental problem.

### 5.3. Analysis of the Material Property Variation in High-Speed

The phenomenon considered in this paper occurs at a velocity of 15 m/s in a short time. Therefore, we carried out a structural analysis of the unsteady state under the assumption of the high physical properties of the material. In general, the stiffness and strength increased at high speed, and therefore the stiffness and breakage strength were set to twice the reference values as given in Section 3.3.

As shown in Figure 11, the support ring was not cut, but rather driven deeper owing to an increase in the urethane foam stiffness. In addition, the maximum deformation of the support ring increased by 18 mm.

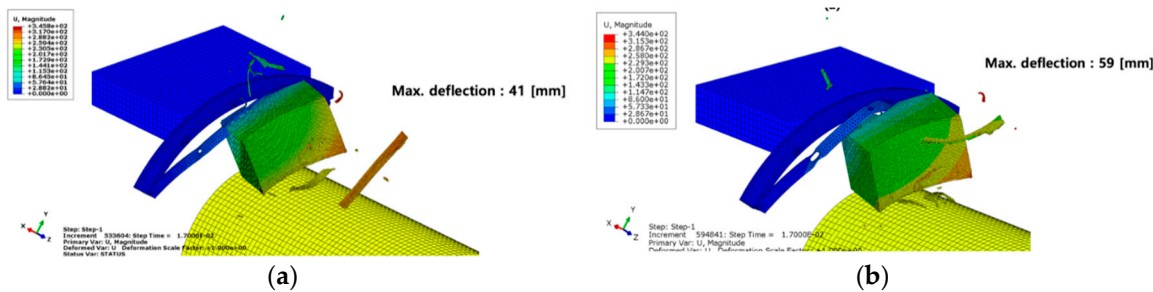

(**a**)　　　　　　　　　　　　　　　　　　　　　　　　(**b**)

**Figure 11.** Effect of the material properties at high velocity (fastener unstable): (**a**) normal and (**b**) high velocity.

### 5.4. Test Result Summary

The overall test results are given in Table 4.

**Table 4.** Summary of the test result.

| | Analysis | Test Result | Impact Factor |
|---|---|---|---|
| Cover Remnant | Deflection of the Support Ring | Increase in the number of bolts reduced the deflection of the support ring. | High |
| | Effect of the Cover Residue Height | Portion protruding beyond the groove height does not significantly affect the deformation of the support ring or behavior of the missile sabot. | Low |
| | Deflection of the Support Ring under the Elimination of Fasteners | Deformed support ring could interfere with the missile booster wing during missile release. | High |
| Material Deterioration | | Urethane deterioration does not cause a fundamental problem. | Low |
| Material Property Variation in High-Speed | | The support ring was driven deeper into the support ring, and the support ring deformation increased. | High |

## 6. Conclusions

The effect of the rupture-type missile canister cover was investigated, while interfering with the neighboring structures, for the case when it is not fully ruptured, and some remnants are present. We also examined the impact of the phenomenon on the ejection performance of a missile by simplifying the shape of the canister cover in the analysis model. A CAE based analysis was then performed and the effects of the cover remnant, material deterioration, and material property variation at high-speed were obtained.

Firstly, no interference between the support ring and missile occurred when the fastening part used for fixing the support ring and frame is normally established. If there is an abnormality in the installment of the fastening part, then the cover residue collides with the missile sabot causing the support ring to deform. From both the simulation and experimental results, we demonstrated that interference can occur with the missile booster.

Secondly, the deterioration of the sabot, which is made of urethane foam, due to long-term storage did not cause any severe problems to the missile launch performance. We confirmed that the urethane foam deteriorates if damaged early; however, it contributes less to the deformation of the support ring.

Thirdly, the material property variation in high-speed resulted in a deeper deformation of the support ring due to an increase in the urethane foam stiffness. When the high-speed properties are reflected, the stiffness is increased and so the missile sabot pulls the support ring for an extended period. This results in an increase in the support ring deformation.

In summary, we concluded that the missile launch performance, for the specific rupture-type missile canister case, could be degraded due to two main causes: (1) the deflection of the support ring after the elimination of fasteners and (2) the material property variation in high-speed.

The simulations can be viewed at the following YouTube URLs: Figure 3 at https://youtu.be/hPQDTxLn58w, Figure 7 at https://youtu.be/A9_ghH_58KY, Figure 8 at https://youtu.be/

BwBMFOYrRWY, Figure 10 at https://youtu.be/eNK05-bXXaw, and Figure 11 at https://youtu.be/dHHzdtEAn0c.

**Author Contributions:** W.C. created models, developed methodology, designed computer programs; S.J. conducted research and investigation process, wrote and edited the initial draft, supervised and leaded responsibility for the research activity planning, reviewed the manuscript, and synthesized study data. All authors read and approved the manuscript.

**Funding:** This research received no external funding.

**Conflicts of Interest:** The authors declare no conflict of interest.

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
