# Peer review of "Launch Performance Degradation of the Rupture-Type Missile Canister"

_applsci, doi:10.3390/app9071290_

Round 1

Reviewer 1 Report

1. The first sentence of the abstract is much too long and incomprehensible.  It should be replaced by short direct sentences. The abstract should tell us what the general topic is, which specific issue is being investigated, the approach used, and what was achieved.  This abstract fails to meet these objectives.

2. The paper describes a finite element analysis of a problem using a commercial code.  It does not identify a research issue that needs to be addressed using new tools or a new approach. 

3. The figures show graphics from the finite element software that do not contain enough information to even identify the various components in the model.

4. There are some serious issues with the English language that should be addressed.

5. This is not a research paper.  It describes a finite element analysis incompletely but does not address any new and significant scientific issue.

Author Response

[Reviewer 1]

The first sentence of the abstract is much too long and incomprehensible.  It should be replaced by short direct sentences. The abstract should tell us what the general topic is, which specific issue is being investigated, the approach used, and what was achieved.  This abstract fails to meet these objectives.

>> As reviewer recommended, the abstract section is modified to be concise.

The paper describes a finite element analysis of a problem using a commercial code.  It does not identify a research issue that needs to be addressed using new tools or a new approach.

>> We used an existing finite element analysis tool to “identify the degradation of launch performance caused by the remnants of a missile canister cover with a sabot interface” which happens frequently in real flight test causing many undesirable emergency situations. As far as we know, none of or scarce researches has been performed to investigate the exact causes of the degradation of launch performance. Additional description is inserted in Section 1 to support our opinion.

The figures show graphics from the finite element software that do not contain enough information to even identify the various components in the model.

>> We can provide more information if you let us know which explanations should be reinforced for better understanding.

There are some serious issues with the English language that should be addressed.

>> We rechecked and modified English language and style of the present paper with professional English proofreading and editing service.

This is not a research paper.  It describes a finite element analysis incompletely but does not address any new and significant scientific issue.

>> Same answer of the second question applies.

Reviewer 2 Report

See attached file

Author Response

[Reviewer 2]

In the reviewed paper, a computational study by Finite-Element Analysis (FEA) is performed to evaluate the effect of the cover remnants in the missile canister structure and missile trajectory. The study is focused to study the role of fasteners, which are installed in the supporting ring, in the structure plastic deformation; and also included some analysis related to the thickness of remnants, degradation of the cover material, and some corrections at high velocities. Additionally, tensile tests in fasteners and visual inspections in missile canisters after real test are also shown.

In general, this paper presents an elaborated FE model of the canister and missile, assuming some valid simplifications due to the axial symmetry of the assembly. Additionally, authors are trying to evaluate (by the FE model) some of the potential parameters that can affect the missile performance. Those two components make this paper a valuable research study. However, many corrections need to be made to be considered as a valid paper for this journal.

1.     First, observation is related to the organization of the paper. According to the directions of the journal, the paper should be organized in the following sections: Introduction, Materials and Methods, Results, Discussion, Conclusions. It was found that the current organization of the paper is confused; where methods, results and discussion are presented at the same time, making the reading confusing and technically not appropriate for a journal paper. Therefore, it is suggested to change the structure of the paper as recommended. Next, the description of the sections, that you can find in the instructions for this journal:

Research Manuscript Sections

·         Introduction: The introduction should briefly place the study in a broad context and highlight why it is important. It should define the purpose of the work and its significance, including specific hypotheses being tested. The current state of the research field should be reviewed carefully and key publications cited. Please highlight controversial and diverging hypotheses when necessary. Finally, briefly mention the main aim of the work and highlight the main conclusions. Keep the introduction comprehensible to scientists working outside the topic of the paper.

·         Materials and Methods: They should be described with sufficient detail to allow others to replicate and build on published results. New methods and protocols should be described in detail while well-established methods can be briefly described and appropriately cited. Give the name and version of any software used and make clear whether computer code used is available. Include any pre-registration codes.

·         Results: Provide a concise and precise description of the experimental results, their interpretation as well as the experimental conclusions that can be drawn.

·         Discussion: Authors should discuss the results and how they can be interpreted in perspective of previous studies and of the working hypotheses. The findings and their implications should be discussed in the broadest context possible and limitations of the work highlighted. Future research directions may also be mentioned. This section may be combined with Results.

·         Conclusions: This section is not mandatory, but can be added to the manuscript if the discussion is unusually long or complex.

>> We reorganize the whole structure again.

2.     In the abstract, it is mention that a parametric study will be performed. A parametric study must to include a define parameter range, to specify the design constraints, and to analyze the results of each parameter variation. Even though that some parameters of the FE model were evaluated, the parametric study is not well implemented, and the results are only qualitatively analyzed. It is expected to have a better description and more discussion related to this parametric study in the paper.

>> As you pointed out, we raised several possible reasons for deteriorating launch performance due to the collision of internal parts in ejection; The height of cover residue, degradation in the urethane form, material property variation at a high speed. In principle, it is more desirable to implement more elaborate parameter study in a reasonable range of factors. However, we observed that the height of cover does not change the degree of deflection on the ring if it beyond the height of support ring where the edges of cover remnants are inserted, which tells us we don’t need to describe all cases in this paper. Also, for the material property in the urethan foam, it was highly difficult to find an appropriate reference presenting the nonlinear behavior perfectly opposed to the case of metals generally well published to the public. Therefore, we carried out only qualitative analysis based on limited information.     

3.     Nomenclature can be omitted, considering that few equations are only mention in the paper.

>> The nomenclature section is removed from the manuscript.

4.     Problem statement could be included in the introduction and in Materials and methods section, as discussed in point 1.

>> We modified contents of Section 2 and renamed as “Assembly Structure of the Missile Canister.”

5.     Figure 1 is taken from a webpage. According to the journal policies, even though that the image was referenced, image files must not be manipulated or adjusted in any way that could lead to misinterpretation of the information provided by the original image. Therefore, it is suggested to avoid this figure, and replace by a sketch of the whole missile system.

>> To help readers understand the whole missile launch system, we decided to leave the picture but not to modify as a reviewer recommended.

6.     Reference in line 105 is wrong.

>> The typo is corrected.

7.     In line 115, it is stated that concentrated forces were calculated by the FE analysis. However, the FE model has not yet introduced in the paper, and some results are already shown (for example the Figure 7). It is suggested to evaluate the organization of the paper, as discussed in point 1

>> We reorganize the whole structure again.

8.     In line 117, it is not clear how the value for Km=0.0002 was chosen. This part needs to be better explained and discussed.

>> You can refer to #6 written by Adewale, In the fig. 5, he described an analytical coefficient of transverse deflection on the annular ring with respect to the circumferential angle in a variation of the ratio of inner diameter to outer diameter. In our case, c=0.93, we can expect that the coefficient has a similar value, Km=0.0002 to the largest case where c=0.7 in his paper. Note that this one is the only paper that describes a theoretical background of the deflection for the clamped-free annular ring under concentrated load, which is analogous to our case.

9.     In line 124, the paper is given results. This information needs to be moved to a Results section, as discussed in point 1.

>> We reorganize the whole structure again.

10.  Impact analysis section could be moved to Materials and Methods section, as discussed in point 1.

>> We reorganize the whole structure again.

11.  In line 166, it is stated that urethane foam properties are obtained from tensile/bending test. However, there is no information related to those test. It could be convenient to show the characterization procedure or a reference (if they were taken from other resource).

>> We obtained the plastic property from the standard specimen test according to ASTM D790-97 with cooperation of certificated material test institution. The photo during the test and s-s curve for a certain specimen described below as an example.

ASTM D790-97

12.  In line 168, it is stated that Johnson Cook model was taken from reference [3]. This reference was not openly found; therefore, it is recommended to explicitly show the materials parameters that were using for the simulation.

>>In the following paper (Determination of Johnson-Cook model constants by measurement of strain rate by optical method written by Stopel, and Dariusz Skibicki, 2016), we have a typical representation of Johnson-Cook model for plasticity. It can be achieved in terms of stress-strain curve in the plastic region using this formula; Here, for simplicity, we assumed that the last temperature term is negligible and estimate the strain curve based on the value of both yield and ultimate stress from ‘Matweb’ website for support ring made of steel, sus304.  

13.  In Table 3, material parameters for Al alloy are not referenced. It is recommended to show the source for this material values.

>> Referred from ‘Matweb’, open to the public material property site (www.matweb.com)

14.  Simulation results (section 5) could be moved to a Results and Discussion section, as discussed in point 1.

>> We reorganize the whole structure again.

15.  In figure 8 (and posted respective video), it can be seen a small plastic deformation of the supporting ring, but not as large as shown in Fig. 9. Please, explain how this deformation (fig. 9) was obtained.

>> Figure 9(a) represents a plastic deformation of the support ring with 10 times scaled-up view in Abaqus for a better comparison but still the same deflection property, around 4.5 [mm]. Also, Figure 9(b) shows the plastic deformation in the visual test after full-scale missile launch test. Unfortunately, the right figure was taken a few months earlier than starting our simulation so it does not have exact measurement data for the deformation. That is why we are only able to compare both results in a qualitative approach.

16.  In line 206, it is stated that the plastic deformation obtained in the simulation proves the analytical results. There is not prove of the analytical result because this is a qualitative comparison. Please verify and comment.

>> The same reasoning above. Since the visual test was implemented before getting into this simulation for identifying the cause of failure of missile test, we did not have a quantitative value of the plastic deformation at that time.

17.  Figure 9b shows results of a real canaster after a real test. It is not mentioned any detail about performed real test, and the conditions of those tests. Please, clarify or comment about this.

>> Just visual inspection after actual full-scale missile launch test on the shipboard launcher before getting into impact analysis described in the paper for the identification of the failure of the missile launch. This was not aimed to correlate the simulation result separately with an elaborate part-level test plan.  

18.  Line 210, reference of Fig. 2 is wrong.

>> The typo is corrected.

19.  Line 221, it is concluded that the deformed supporting ring might interfere with the booster wing. This is a subjective observation and any evidence in the simulations clarifies this statement. Please clarify and comment about this.

>> The payload has folded wings in boosters located in the rear side of it, when the wings start to open up, the overall height beyond the outer diameter of the canister. In this end, it is highly likely to interfere with the booster wing if even small detachment of the support ring occurs.

20.  Line 227, reference of Fig. 2 is wrong

>> The typo is corrected.

21.  Between line 225 and 247, a tensile experiment in the fasteners is presented to validate resistance of the ring. This experiment must to be better explain and more information should be given to understand the comparison between the experiment and the real case. For example: are the same materials used in the tensile test and the ring and fasteners of the canister? Is the same type of fasteners? Is the experiment valid for the real case (strain-rates, stress conditions, etc..) ?

>> We used the exact same kind of fastener and a mating part, so-called ‘insert’ that strengthen the assembly through the hole on the base part as well as the same material for the support ring and base aluminum frame. The purpose of this experiment was to verify the influence of the deviation of clearance from the nominal fit between an insert and hole on the base frame on the maximum endurable tensile force under a general tensile test. From the results, we observed that the faster is supposed to be broken at higher tensile force above 1,000 [kgf] in a normal fit condition for the insert as opposed to the fact that the insert itself is pulled out with the fastener in a bigger clearance condition, which tells us the possibility the detachment of insert with the faster at relatively low tensile force from 100~590 [kgf] under the mal-fabrication of clearance between the insert and hole of base frame.

22.  In line 253, Chapter 3 is not a section of this paper.

>> It is edited as “Section III-B.”

23.  For the urethane deterioration study, the change in properties is based on subjective arguments. It is recommended to varied the parameters and make a deeper parametric study.

>> The same reason as 24 below.

24.  In the high-speed properties study, it is not valid to only change the properties of the materials for the simulation. Instead, a proper plastic model should be used, which includes a strain-rate dependent term to account velocity changes.

>> There is no theoretical reference on strain-stress dependent plastic model for urethane form as far as we found. We can obtain those only from the experimental data but unfortunately do not have a database at high-speed circumstances.

25.  In the cover residual height study, it is recommended to perform a deeper parametric study of this parameter.

>> In principle, it is more desirable to implement more elaborate parameter study in a reasonable range of factors. However, we observed that the height of cover does not change the degree of deflection on the ring if it beyond the height of support ring where the edges of cover remnants are inserted by simulation already, which tells us we don’t need to describe all cases in this paper.

After this evaluation, it is recommended for this paper to Reconsider after Major Revisions.

Reviewer 3 Report

Major issues:

1. Overall lack of consistency of units. SI(N,   mm..), British units (inch, psi) and even non-standard SI units (kgf). Inconsistent even within SI units like GPa, mm. Inconsistent italics in units, mix of regular and italic units. (N, N)

2. Line 210, use labels instead of #1, #2 and so on. Nobody wants to hunt the paper for what’s #1, #2 and so on. Throughout the article as well, especially Figure 12.

3. Line 155, provide reference for this logical jump. From the tests the dynamic properties of steel doubled, so its logical to assume that urethane foam properties will also increase but its unsubstantiated without proof. Let alone why two fold increase similar to steel?

4. Section 5.2 “Simulate urethane deterioration” and utilizing deteriorated material properties are completely different aspects.

5. Given that the fasteners are of the same size used, why the maximum tensile force for the same fasters is 400% more?

6. Section 5.3 Consider using technically sound terms instead of “High-Speed property”, let alone on section title.

7. Line 297, It’s not “analytically” proven. Consider revising this sentence.

Minor issues:

8. Line 93, “cover remnants are left between support ring” and …? Sentence is incomplete.

9. Figure 4, Title: It’s  not “analytical model” (Its incorrect or Wrong word choice at best). Not only here, this seems to be present across the article.

10. Fig 7 title, “Numerical result on a deflection” ?

11. Line 134, be clear with you mean “upper region”

12. Line 142, “showed investigated” next each other will never work in a sentence.

13. Line 144, “maximum stress” meaning tensile strength?

14. Line 149, “high-speed circumference” ?

15. Line 153, “s-s curve”? Be clear, stress-strain curve?

16. Line 182, “Mutual speed”? velocity you mean?

17. Line 186, “To determine”, It’s not determining

18. Line 189, “element deletion” for which one? Urethane foam?

19. Figure 8, Legends and title block at the bottom of each picture in the figure does nothing other than obscuring things. Instead of beginning, middle and end consider using time. Applies to all the relevant pictures.

20. Figure 9, Instead of U, either PE or PEEQ might      be better.

21. Line 236, “actual” doesn’t signify anything.

22. Line 265, “fast speed”?

23. Line 298, “clamping part”?

Author Response

[Reviewer 3]

Major issues:

Overall lack of consistency of units. SI(N, mm..), British units (inch, psi) and even non-standard SI units (kgf). Inconsistent even within SI units like GPa, mm. Inconsistent italics in units, mix of regular and italic units. (N, N)

>> All units throughout the presented paper are corrected in SI units.

1.     Line 210, use labels instead of #1, #2 and so on. Nobody wants to hunt the paper for what’s #1, #2 and so on. Throughout the article as well, especially Figure 12.

>> As the reviewer suggested, we enhanced by adding exact figure numbering for the readability.

2.     Line 155, provide a reference for this logical jump. From the tests the dynamic properties of steel doubled, so it’s logical to assume that urethane foam properties will also increase but its      unsubstantiated without proof. Let alone why two fold increase similar to steel?

>> We could not find any appropriate reference paper on the variation of a property of urethane form at high speed.

3.     Section 5.2 “Simulate urethane deterioration” and utilizing deteriorated material properties are completely different aspects.

>> In our cause analysis, urethane deterioration was considered to be one of the reasons for the collision between the sabot and support ring. To do this, we speculated the material degradation of urethane form mostly based on the papers assisted with this. According to the normal knowledge, both strength and stress are decreased by the long-term storage caused by temperature change, ultraviolet light, and humidity etc. Moreover, I would like to be informed more details of what you wish to be explained in terms of simulating urethane deterioration.

4.     Given that the fasteners are of the same size used, why the maximum tensile force for the same fasters is 400% more?

>> Basically, the fasters is assembled with so-called ‘insert’ component that is already installed on the hole of base frame for protecting the thread of fasteners under longer usage in the field. However, the insert is not installed with the designated diameter of the hole on the base frame, the insert would be pulled out with the fasteners from the hole, which decreases the endurable tensile force. That is why it has a variation above 400% more depending on the deviation of hole size with respect to the fasteners. Otherwise, the only fasters might be broken without the detachment of insert at a high tensile force of the fasters.

5.     Section 5.3 Consider using technically sound terms instead of “High-Speed property”, let alone on section title.

>> We could not find appropriate alternative term instead of “high-speed property” and so kept as it is. If the reviewer has any good idea, please share it with us.

6.     Line 297, It’s not “analytically” proven.  Consider revising this sentence.

>> We removed the term, “analytically.”

Minor issues:

1.     Line 93, “cover remnants are left between support ring” and …? Sentence is incomplete.

>> We modified the sentence as “When the missile canister cover is not completely removed by the ignition shock wave before launch, the cover remnants are left between the support ring and canister and it eventually presses and fixes the rupture-type canister cover.”

2.     Figure 4, Title: It’s not “analytical model” (Its incorrect or Wrong word choice at best). Not only here, this seems to be present across the article.

>> We removed the term “analytical” throughout the paper.

3.     Fig 7 title, “Numerical result on a deflection” ?

>> We modified the sentence as “Deflection of the support ring (fastener stable).”

4.     Line 134, be clear with you mean “upper region”

>> We removed the sentence.

5.     Line 142, “showed investigated” next each other will never work in a sentence.

>> We modified the typo.

6.     Line 144, “maximum stress” meaning tensile strength?

>> Yes, we modified the sentence as “Tcharkhtchi [10] investigated the effect of thermo-aging on the urethane foam using Young's modulus and the maximum stress of tensile tests based on chemical reaction after 62 weeks aging at 85°C.”

7.     Line 149, “high-speed circumference” ?

>> We replaced the term with “high-speed environment.”

8.     Line 153, “s-s curve”? Be clear, stress-strain curve?

>> Yes, we modified the term as “stress-strain curve.”

9.     Line 182, “Mutual speed”? velocity you mean?

>> Yes, we modified the term as “velocity.”

10.  Line 186, “To determine”, It’s not determining

>> We modified the sentence as “We employed the general contact function, which is a default option in Abaqus 6.141, to set the type of contact between components.”

11.  Line 189, “element deletion” for which one? Urethane foam?

>> We modified the sentence as “We also employed special feature of element deletion for Urethane foram supported by Abaqus 6.141, in which the element is removed at a plastic strain of 6% or greater, to analyze the fracture behavior at the contact surface.”

12.  Figure 8, Legends and title block at the bottom of each picture in the figure does nothing other than obscuring things. Instead of beginning, middle and end consider using time. Applies to all      the relevant pictures.

>> We could not perform additional Abaqus simulations due to software license expiration. Instead, we modified the figure size.

13.  Figure 9, Instead of U, either PE or PEEQ might be better.

>> We could not perform additional Abaqus simulations due to software license expiration. Instead, we modified the figure size.

14.  Line 236, “actual” doesn’t signify anything.

>> We removed the whole description regarding the real tensile strength test from the manuscript.

15.  Line 265, “fast speed”?

>> We replaced the term with “velocity.”

16.  Line 298, “clamping part”?

>> We removed the whole description regarding the real tensile strength test from the manuscript.

Round 2

Reviewer 1 Report

In the abstract, the use of the word interference is confusing.  What are we talking about: impact?

On p.2, “Johns–Cook” is misspelled

Figure 1 is said to show several things: canister, cover assembly,… For the benefit of those not familiar with this, it would be nice to have arrows pointing to those parts so we know exactly what we are talking about.

In Figs. 3, 7, 8, 9, 10, and 11, the legends are not legible and it is not possible to get any quantitative information.

The conclusion does not tell us what can be considered to be a new and significant contribution. It does not refer to the existing literature and does tell us what this paper offers that has cannot be found in existing publications.

While the revised version is an improvement over the initial version, the main issue remains.  What is the significant scientific issue that is addressed in this manuscript?  The argument made in the author’s reply is that there is none but that there are few publications on this particular mechanical device.  Without challenging the quality of the work, research papers should address specific scientific issues and present significant advances in dealing with those issues.
